# Functional Mitral and Tricuspid Regurgitation across the Whole Spectrum of Left Ventricular Ejection Fraction: Recognizing the Elephant in the Room of Heart Failure

**DOI:** 10.3390/jcm12093316

**Published:** 2023-05-06

**Authors:** Valeria Cammalleri, Giorgio Antonelli, Valeria Maria De Luca, Myriam Carpenito, Annunziata Nusca, Maria Caterina Bono, Simona Mega, Gian Paolo Ussia, Francesco Grigioni

**Affiliations:** Research Unit of Cardiovascular Science, Università e Fondazione Policlinico Universitario Campus Bio-Medico, Via Alvaro del Portillo, 200, 00128 Roma, Italy

**Keywords:** heart failure, mitral regurgitation, tricuspid regurgitation, atrial functional mitral regurgitation, atrial functional tricuspid regurgitation, transcatheter tricuspid valve intervention

## Abstract

Functional mitral regurgitation (FMR) and tricuspid regurgitation (FTR) occur due to cardiac remodeling in the presence of structurally normal valve apparatus. Two main mechanisms are involved, distinguishing an atrial functional form (when annulus dilatation is predominant) and a ventricular form (when ventricular remodeling and dysfunction predominate). Both affect the prognosis of patients with heart failure (HF) across the entire spectrum of left ventricle ejection fraction (LVEF), including preserved (HFpEF), mildly reduced (HFmrEF), or reduced (HFrEF). Currently, data on the management of functional valve regurgitation in the various HF phenotypes are limited. This review summarizes the epidemiology, pathophysiology, and treatment of FMR and FTR within the different patterns of HF, as defined by LVEF.

## 1. Introduction

Functional (secondary) mitral (FMR) and tricuspid (FTR) valve regurgitation are shared across the entire spectrum of heart failure (HF) and negatively affect symptoms and prognosis [1,2]. They may occur isolated or concomitantly (bivalvular functional regurgitation), independent of the HF subgroup [3]. By definition, any functional regurgitation occurs due to cardiac remodeling and dysfunction and appears in a structurally normal valve apparatus [3,4,5,6,7]. Annular dilatation and impaired contraction cause atrial functional regurgitation. Restricted motion of the leaflets due to ventricular remodeling and dysfunction produces ventricular functional regurgitation. We can diagnose FMR and FTR in any HF phenotype as defined by left ventricular ejection fraction (LVEF): preserved (HFpEF), mildly reduced (HFmrEF), or reduced (HFrEF). Proper and simultaneous recognition of the specific mechanism of regurgitation on the one hand (functional atrial, ventricular, or mixed) and the phenotype of HF on the other (HfrEF, HFmrEF, and HFpEF) is crucial for prognosis and therapy. In the present review, we aim to focus on the epidemiology, pathophysiology, prognosis, and therapy of atrial and ventricular FMR and FTR within the different HF phenotypes defined by LVEF.

## 2. Epidemiology

In HF, moderate or severe FMR affects up to 30% of patients, and it seems more frequent in HFrEF, followed by HFmrEF and HFpEF [2]. The prospective analysis of the European Society of Cardiology (ESC) Heart Failure Long-Term Registry shows a prevalence of moderate-to-severe FMR approaching 35% in the HFrEF group, 30% in the HFmrEF group, and 20% in the HFpEF group (*p* < 0.001) (Figure 1) [8]. In advanced HFrEF (stage C–D), the prevalence of severe FMR can reach 45% [9,10,11,12]. There are no dedicated studies linking the prevalence of the specific mechanism causing FMR (atrial vs. ventricular) to the single HF phenotypes (HFrEF, HFmrEF, and HFpEF). However, we can hypothesize that in HFrEF, ventricular mechanisms are likely to prevail, but atrial mechanisms can coexist and are proportional to the disease severity. Moving from HFrEF to HFmrEF and HFpEF, the ventricular mechanisms become less relevant, leaving atrial mechanisms the primary determinants of FMR.

Most studies on FTR focus on the community and not specifically on HF [13,14,15]. An incidental finding of moderate and severe FTR occurs in 7% of the general population and 12% of patients hospitalized for HF [15,16,17]. The ESC Heart Failure Long-Term Registry reports a prevalence for moderate-to-severe FTR equally distributed among HF phenotypes, ranging from 18% in HFmrEF to 20% in HFrEF and 21% in HFpEF (*p* = 0.164) (Figure 1) [8,18]. Since no data on right atrial and ventricular remodeling are available in this study, it is impossible to establish the role of the atrial and ventricular mechanisms of FTR across HF phenotypes. The significantly older age of HFpEF patients does not allow for excluding a coexisting organic etiology in these patients [14].

In the entire spectrum of HF, FMR and FTR often coexist. Moderate or severe bivalvular functional regurgitation has been observed in about 35% of patients suffering from HFrEF [1,18]. In the same way, biatrial dilatation is commonly present in patients with HFpEF, resulting in concomitant aFMR and aFTR [19]. Significant bivalvular functional regurgitation is rarely observed in patients with sinus rhythm or atrial fibrillation (AFib) ≤1 year. In contrast, 25% of patients with AFib >10 years have significant bivalvular regurgitation, adding complexity to diagnosis and management [19].

## 3. Pathophysiology and Prognosis

Two main mechanisms are responsible for functional mitral and tricuspid regurgitation: (1) the annular dilation and/or loss of annular contraction, through a condition of atrial remodeling (atrial functional); (2) restricted leaflets motion due to ventricular remodeling, which implies papillary muscle displacement, causing chordal tethering (ventricular functional). These geometrical alterations and functional impairments occur in the presence of a structurally normal valve apparatus.

Ventricular FMR typically occurs in HFrEF due to ischemic or non-ischemic ventricular disease. According to the general classification, the presence of coronary artery disease affecting LV geometry and function allows for differentiation between ischemic and non-ischemic FMR. Dilated cardiomyopathy (DCM), regardless of its etiology, often leads to secondary MR, due to the changes in LV shape (increase in LV sphericity and enlargement in LV diameters). DCM recognizes genetic, but also acquired causes. Monogenic diseases, syndromic forms, and neuromuscular diseases are described among genetic forms. Drugs, toxins, and nutritional deficiencies can lead to acquired forms of DCM with FMR.

The mechanism of FMR is valve tenting (a more apical position of the leaflets and their coaptation point during the systolic phase) (Figure 2). Specifically, valve tenting results from an imbalance between tethering and closing forces. In ventricular FMR, tethering forces increase (due to LV remodeling), and closing forces decrease (due to reduced contractility and dyssynchrony) [20,21]. Valve tenting can be symmetric or asymmetric. While symmetric tenting occurs more often in global ventricular remodeling, asymmetric tenting usually occurs if the tethering forces predominate on the posterior mitral valve leaflet. FMR negatively impacts survival, either in HFpEF [adjusted hazard ratio (adj. HR) 1.40, 95% confidence interval (CI) 1.09–1.81; *p* = 0.009] [22], HFmrEF (adj. HR 1.72, 95% CI 1.24–2.39; *p* = 0.0012) [8] and HFrEF (adj. HR 1.61, 95% CI 1.22–2.12; *p* = 0.001] [2]. In HFrEF, small amounts of FMR increase short- and long-term mortality. Particularly, there is an exponential mortality increase for any effective regurgitant orifice area (EROA) increment above a threshold of 0.10 cm^2^ when compared with degenerative MR [23].

Atrial FMR is common in AFib but also occurs in sinus rhythm. HFpEF can generate atrial FMR by causing an increase in left atrium (LA) pressure and, eventually, LA remodeling without needing AFib to develop (Figure 2) [6,7,19,24,25].

Previously published data showed that not all patients with significant aFMR had known atrial arrhythmias. Dziadzko V. et al. found that 46% of patients with aFMR do not have atrial arrhythmias [24]. More recently, Mesi O. et al. demonstrated that 23% of the aFMR population had sinus rhythm [26]. This suggests that diastolic dysfunction with resultant atrial dilation and annular remodeling could be sufficient in promoting the genesis of mitral regurgitation. Nevertheless, AFib, HFpEF, and atrial FMR often coexist and negatively interact since they share most pathophysiological mechanisms [26,27,28,29,30,31]. AFib, causing LA remodeling, impaired atrial function, and atrial fibrosis, may negatively contribute to HFpEF and atrial FMR [28,29,30,31,32]. HFpEF, through diastolic dysfunction and increased LA pressures, systemic inflammation, and endothelial dysfunction, plays a crucial role in causing LA anatomical, mechanical and electrical remodeling favoring AFib and, consequently, atrial FMR. Once established, FMR negatively contributes to AFib and HFpEF progression.

Figure 3 resumes the complex pathophysiological relationship between AFib, LA enlargement, and MR.

In a Dziadzko V et al. study, patients with aFMR were significantly older than those with vFMR (80 ± 10 vs. 73 ± 14 years), translating into a different distribution of causes by age group [24]. The aFMR patients suffered mainly from atrial fibrillation/flutter (54% vs. 28%) and hypertension (81% vs. 69%). In contrast, vFMR patients were predominantly male (59% vs. 33%) with a prevalent history of myocardial infarction (17% vs. 9%) [24]. In addition, patients with ventricular FMR had the most significant LV remodeling, highest pulmonary pressure and lowest LVEF, stroke volume, and E/e’. Patients with atrial FMR presented smaller LV size, generally normal LVEF and stroke volume, with a modest MR volume and orifice, while E/e’ and pulmonary pressure were elevated [24]. In advanced LA and LV remodeling, a net distinction between the atrial and ventricular mechanism is no longer possible because these entities usually coexist. In HFmrEF, the volume overload caused by atrial FMR promotes the transition to HFrEF (and eventually to ventricular FMR) [18].

Even if current guidelines do not emphasize the need to discriminate the atrial from the ventricular mechanism in FMR, an early distinction is crucial to establish prognostic and therapeutic decisions [19]. The prognosis of ventricular FMR is significantly worse than atrial FMR, and each etiology leads to different treatments [24,33]. Though, the question remains whether the relationship between vFMR and mortality is direct or indirect, assuming that FMR is independently responsible for the outcomes and in all circumstances. On the one hand, a direct relationship between the degree of FMR and mortality has been widely described; on the other hand, several cohort publications stated that FMR was not independently responsible for the poor outcomes observed, suggesting that FMR is a surrogate for another cause of reduced survival [24,33,34]. In very advanced HFrEF, the underlying myocardial impairment and severity of LV dysfunction have a more negative impact on prognosis than FMR [18].

Similar to the left side of the heart, right ventricular remodeling, causing leaflet tethering and systolic restricted motion, is typical of vFTR. This can occur in case of left heart diseases (left ventricular dysfunction or left heart valve diseases) resulting in pulmonary hypertension, primary pulmonary hypertension, secondary pulmonary hypertension and right ventricular dysfunction from any cause (e.g., myocardial diseases, ischemic heart disease, chronic right ventricular pacing). Atrial FTR develops due to tricuspid annular dilatation following right atrium (RA) remodeling, with the concomitant valve leaflets, right ventricle (RV), pulmonary circulation, and left side of the heart being macroscopically normal (Figure 4) [35,36,37,38]. In HFpEF, due to cardiac amyloidosis complicated by atrial FTR, an organic component usually coexists because of amyloid deposit infiltration in the leaflets [39,40].

A remarkable past medical history for AFib is widespread in atrial FTR. Atrial and ventricular FTR can coexist in simultaneous RA and RV remodeling. The same happens in FTR due to cardiovascular implantable electronic devices [37].

A stand-alone diagnosis of atrial FTR should make us search for HFpEF [41]. The high prevalence of atrial FTR in HFpEF is consistent with shared risk factors such as renal dysfunction, aging, and AFib. AFib is also a primary determinant of atrial FTR. In HFrEF, the role of AFib in determining FTR diminishes. Compared to HFpEF, a lower percentage of patients with HFrEF have AFib [16,42,43]. In HFrEF, right ventricular remodeling and dysfunction are the main determinants of ventricular FTR.

Distinguishing between the atrial and ventricular FTR has prognostic and therapeutic implications [44,45,46]. The presence of FTR in the HF population significantly impairs prognosis, functional capacity, and quality of life and increases the risk of hospital admission. A strong association between FTR and mortality exists both in HFrEF (adj. HR 1.30, 95% CI 1.06–1.60; *p* = 0.014) [47] and HFpEF (adj. HR 2.87, 95% CI 1.61–5.09; *p* < 0.001) [48]. To our knowledge, dedicated studies on HFmrEF are missing, but the presence of FTR is proven to be an independent risk predictor of mortality in mixed cohorts of HFrEF and HFmrEF patients (adj. HR 1.57, 95% CI 1.39–1.78; *p* < 0.0001) [16,49]. Atrial FTR progresses rapidly but has a better outcome than ventricular FTR [37,50]. Additionally, while regurgitation severity is the only independent prognostic predictor in atrial FTR, RV function also predicts outcomes in ventricular FTR [16,50,51,52].

## 4. Therapeutic Implications

A multidisciplinary approach is a cornerstone for adequately managing HF complicated by FMR or FTR. The team should include HF specialists, imaging experts, cardiac surgeons, interventional cardiologists, and electrophysiologists. Proper management of comorbidities, such as hypertension, diabetes, renal dysfunction, and depression, is also essential and improves outcomes [53,54,55,56].

The first therapeutic approach includes guideline-directed drug therapy (GDMT), followed by surgical valve correction when indicated. Transcatheter repair and replacement for FMR and FTR are emerging as complementary and promising therapeutic options across all HF phenotypes. These techniques can significantly reduce the harmful effects of regurgitant volume overload and interrupt the vicious circle of valvular-driven HF progression.

### 4.1. Functional Mitral Regurgitation

GDMT is the first mandatory therapeutic step in FMR complicating HFrEF (and likely HFmrEF and HFpEF). Treatment with beta-blockers, renin–angiotensin–aldosterone system antagonists, angiotensin receptor neprilysin inhibitors, and most recently, sodium-glucose co-transporter inhibitors, which may result in LV unloading and reverse remodeling and pleiotropic drug effects, secondarily reducing FMR [18,38,57]. Following GDMT, appropriately selected patients can take advantage of cardiac resynchronization therapy (CRT) [38,57]. Medical therapy and CRT can improve atrial and ventricular FMR by favorably acting on leaflet tethering and closing forces and ventricular performance and decreasing LA pressure. Bartko et al. found that the interpapillary longitudinal dyssynchrony was markedly increased in patients with severe FMR than moderate or less FMR. Restoration of longitudinal papillary muscle synchronicity by CRT was correlated with FMR regression. Similarly, the improvement of FMR was associated with improved interpapillary radial and longitudinal dyssynchrony [58]. Unfortunately, the positive effects of CRT are not immediate, and only about half of the patients implanted take advantage of it [18,54]. A positive response to CRT implantation is expected in the presence of an anteroseptal to posterior wall radial strain dyssynchrony > 200 milliseconds and an end-systolic LV dimension indexed < 29 mm/m^2^, and in the absence of a scar at lead insertion [59]. In addition, FMR improvement after CRT is less common in patients with AFib than sinus rhythm despite a comparable extent of LV reverse remodeling [58,59,60]. Herein, restoring sinus rhythm before CRT implantation may positively affect the time course of FMR severity [25,53]. Reestablishing sinus rhythm, regardless of the LV function, has a therapeutic effect by reversing LA anatomical and mechanical remodeling, particularly on atrial FMR. Dell’Era et al. observed a significant improvement in the LA deformation index (peak atrial longitudinal strain), LA volume, and FMR grade shortly after cardioversion [61]. Gertz et al. reported that successful catheter ablation for AFib results in a significant reduction in LA size and annular dimension and lower rates of important atrial FMR [25]. Taken together, these data, on the one hand, highlight the role of AFib (and LA and annular remodeling) in causing atrial FMR, and on the other, they provide therapeutic indications for its treatment [38,57,62].

Surgical or transcatheter treatment is an option in patients with persistent FMR despite GDMT and, when applicable, CRT [38,57]. Nowadays, an isolated surgical approach to ventricular FMR is rare because of the considerable risk of surgery and the remarkable recurrence rates after mitral repair in the presence of LV remodeling [56,57]. On the contrary, when the primary mechanism of FMR is annular dilation (atrial FMR), a surgical approach targeting the mitral annulus only is a valuable option. The results of surgical annuloplasty for atrial FMR are encouraging [62,63], although this approach, when isolated, is not always sufficient [24,25,48]. In this scenario, transcatheter therapies for FMR, thanks to their potentially low procedural risks and long-lasting results, are the focus of intense clinical research in atrial and ventricular FMR settings. 

Table 1 summarizes the most applied transcatheter techniques currently commercialized for managing MR.

European guidelines recommend transcatheter edge-to-edge repair (TEER) in high-risk symptomatic MR not eligible for surgery and satisfying a set of anatomic criteria. This recommendation applies both to functional (Class IIa, level of evidence B) and degenerative (Class IIb, level of evidence B) etiology [57]. American guidelines recommend TEER in chronic severe FMR and persistent symptoms despite GDMT in patients fulfilling specific anatomical criteria: LVEF 20–50%, left ventricular end-systolic dimension ≤ 70 mm, and pulmonary artery systolic pressure ≤ 70 mmHg (Class IIa, level of evidence B). The same evidence of recommendation is applicable for degenerative MR in patients with high or prohibitive surgical risk if mitral valve anatomy is favorable [38].

The Cardiovascular Outcomes Assessment of the MitraClip Percutaneous Therapy for Heart Failure Patients with Functional Mitral Regurgitation (COAPT) and Percutaneous Repair with the MitraClip Device for Severe Functional/Secondary Mitral Regurgitation (MITRA-FR) trials compared the TEER with the MitraClip device vs. GDMT in patients with FMR [66,67]. In the COAPT trial, MitraClip was superior to medical therapy alone at two years in reducing mortality and rehospitalization [66]. In the MITRA-FR trial, the mortality and the rehospitalization rate at one year were similar in the two arms of treatment [67]. A comprehensive analysis of the discrepancies between these two studies led to a complete knowledge of the patients enrolled and their echocardiographic characteristics (Table 2). Grayburn et al. proposed a new conceptional framework reconciling the results of the MITRA-FR and COAPT, based on the concordance between the grade of FMR and the amount of LV dilatation (“proportional” MITRA-FR-like) [68]. The authors concluded that, in “proportional“ patients, MR correction would bring little or no improvement to a diseased ventricle affected by a nonsignificant amount of FMR-induced volume overload. Conversely, the benefit might be higher with a relatively large EROA associated with only a moderately dilated ventricle (“disproportionate” COAPT-like). Although this concept is attractive and elegant from an intellectual point of view, it seems to be mainly a theoretical assumption, because the echocardiographic characterization of MR and the hemodynamics of the patients in both studies are not convincing [69]. In order to assess the FMR proportionality, some authors suggest focusing on the ratio of regurgitant volume to ventricular end-diastolic volume. This ratio could anticipate the extent of reverse remodeling occurring after FMR correction. Similarly, regurgitant fraction, by including FMR severity, ventricular volumes and function may also provide prognostic information [69]. Interestingly the rate of AFib and/or atrial flutter was 34.5% in the MITRA-FR trial vs. 57.3% in the COAPT trial, suggesting also different MR etiologies among enrolled patients. Most recently, the single-arm Transcatheter Mitral Valve Repair System Study (CLASP) assessed the PASCAL edge-to-edge device (Edwards Lifesciences, Irvine, CA, USA) in patients with degenerative and functional MR [70]. In the FMR cohort, the two-year mortality and freedom from HF hospitalizations were comparable with that obtained from the MitraClip in the COAPT trial (CLASP: 28% and 78%, respectively; COAPT: 30% and 65%, respectively). In addition, two years after treatment, the prevalence of significant MR decreased and symptoms improved, as confirmed by 95% of patients with MR ≤ moderate and 88% of patients in New York Heart Association (NYHA) functional class I–II after the procedure as compared to 36% at baseline [70]. Table 2 summarizes the main difference between COAPT, MITRA-FR and CLASP studies.

A recent study by Gertz ZM et al. analyzed patients in the COAPT trial with a history of AFib, assuming they were most likely to have a mixed atrial and ventricular FMR [71]. Patients with a history of AFib had larger LA, higher LVEF, smaller LV volumes, and similar FMR severity. Patients with a history of AFib had a worse prognosis but benefited from the MitraClip [71]. Other studies investigating TEER in atrial FMR confirmed that this approach could provide sustained FMR reduction over two years and improved clinical outcomes [72,73,74]. In particular, LA volume index and leaflet-to-annulus index may predict the extent of improvement of atrial FMR provided by TEER [75].

Transcatheter approaches mimicking surgical annuloplasty could also be helpful, particularly in atrial FMR [75,76,77].

Although preliminary data from multiple transcatheter techniques are encouraging, further studies are warranted to determine the most appropriate strategy for the different phenotypes of FMR across the entire LVEF spectrum.

### 4.2. Functional Tricuspid Regurgitation

FTR in HFrEF, by increasing the risk of overt right-sided HF and end-organ dysfunction, negatively impacts symptoms and prognosis. Although consistent data on the direct effect of GDMT on the reduction of TR are missing, diuretics, sodium, and water restriction by acting on volume overload remain the cornerstone of medical treatment [78].

Scientific guidelines recommend surgery for severe FTR only in the presence of associated left-sided lesions deserving simultaneous treatment. In these patients, surgery for FTR is also an option when the degree is not severe but the annulus is dilated [38,57]..

Depending on the predominant etiology, surgical corrective measures should restore valve competence by addressing the underlying specific mechanisms. When annular dilatation is the primary mechanism of FTR (atrial FTR), surgical annuloplasty is the preferred approach to reduce annular dimensions, remodel annular shape, and improve leaflets coaptation [38,57]. On the contrary, surgical annuloplasty carries a high risk of recurrence in the case of ventricular FTR because of the significant leaflet tethering and RV dysfunction/remodeling [79,80]. In routine practice, isolated tricuspid valve surgery, particularly valve replacement, is rare because of the significant operative risk, mainly linked to the high prevalence of severe comorbidities in these patients [81]. Accordingly, the scientific community now perceives transcatheter tricuspid valve interventions as a potential tool to improve symptoms and perhaps the prognosis of patients with HF complicated by FTR.

Table 3 summarizes the most applied transcatheter techniques currently used to manage FTR.

The TriValve Registry showed a procedural success rate of 73%, periprocedural mortality of 0%, and a 30-days adverse event rate of 11% [84]. A sub-optimal result of the procedure, defined as residual regurgitation ≥ grade 2+, was a predictor of future adverse outcomes [84]. Specifically, predictors of suboptimal procedural results were baseline TR grade (defined by EROA), gap of coaptation, tenting area, and TR jet localization (no central or anteroseptal). Additionally, clinical predictors of one-year mortality included procedural failure, worsening renal function, and absence of sinus rhythm [84]. A successive propensity-matched analysis compared the transcatheter valve therapy of the TRiValve population to medical treatment alone. The results suggest that transcatheter intervention is associated with more favorable one-year survival and freedom from HF hospitalizations even after the adjustment for confounders at baseline [85]. In the transcatheter plus medical therapy group, 22% of patients had LVEF ≤ 35% (21% in the medical therapy alone group). This figure indicates that most patients enrolled in the registry were likely affected by HFmrEF and HFpEF complicated by atrial FTR. Nevertheless, the transcatheter treatment improved the outcome in all subsets of LVEF [85].

Recent data from the randomized TRILUMIATE trial showed that TEER with the TriClip system (Abbott Vascular, Abbott Park, IL, USA) was safe for patients with severe TR, reducing the grade of regurgitation, and improving the quality of life [86]. Into details, the primary end point (including death from any cause or tricuspid-valve surgery; hospitalization for HF; and an improvement in quality of life) favored the transcatheter group over medical controls (win ratio, 1.48; 95% confidence interval, 1.06 to 2.13; *p* = 0.02). At 30 days, 87.0% of the TriClip patients and 4.8% of medical therapy patients had TR of no greater than moderate severity (*p* < 0.001). In addition, the quality-of-life score changed by a mean (±SD) of 12.3 ± 1.8 points in the TriClip group, as compared with 0.6 ± 1.8 points in the control group (*p* < 0.001) [86]. As regards the echocardiographic characteristics of the patients enrolled, 94.8% of the TEER group and 92.9% of the control group had FTR; the mean LVEF was 59.3 ± 9.3% in the TEER group and 58.7 ± 10.5% in the control group, with 14% of patients with LVEF < 50% in both groups. In the TriClip group, 87.4% and 11.4% suffered from AFib and atrial flutter, respectively; in the medical therapy group, 92.6% and 12.6% had AFib and atrial flutter, respectively. These data suggest that most patients enrolled in the trial were likely affected by aFTR, associated with a preserved or mildly reduced LVEF.

Although surgery can effectively address atrial FTR, most patients with HFmrEF and HFpEF remain untreated because they are considered higher-risk surgical candidates. However, data from relatively small experiences and registries suggest that a transcatheter approach is valuable even for these patients [40,86,87,88]. The correction of atrial FTR while reducing right atrial pressure, backward signs, and symptoms of right HF driven by congestion could increase forward stroke volume [40]. The increase in cardiac output could have a clinical and prognostic impact, particularly in patients with restrictive pathophysiology, such as HFpEF [40]. Therefore, transcatheter annuloplasty could be effective when leaflet tethering is less pronounced, and TR is mainly due to RA and tricuspid annulus dilatation (atrial FTR) [82,86,89]. In patients with advanced geometrical remodeling, TV replacement could be a promising option that is still in its infancy, but no transcatheter devices are currently available for commercial use [37,89].

Prospective randomized studies addressing these unmet clinical needs across all HF stages must confirm these hypotheses generated from exploratory studies.

## 5. Conclusions

Functional MR and TR are common findings in HFrEF, HFmrEF, and HFpEF. The proper and simultaneous recognition of the specific mechanism of regurgitation on the one hand and the phenotype of HF on the other is crucial for defining prognosis and therapy. GMDT is the first-line treatment for functional regurgitation across all HF phenotypes, followed by CRT in appropriately selected patients. Behind GDMT and CRT, surgical or transcatheter valve therapy is a valuable option for patients remaining symptomatic. Pharmacological and non-pharmacological treatments are complementary and can interrupt valvular-driven HF progression in appropriately selected patients.

## Figures and Tables

**Figure 1 jcm-12-03316-f001:**
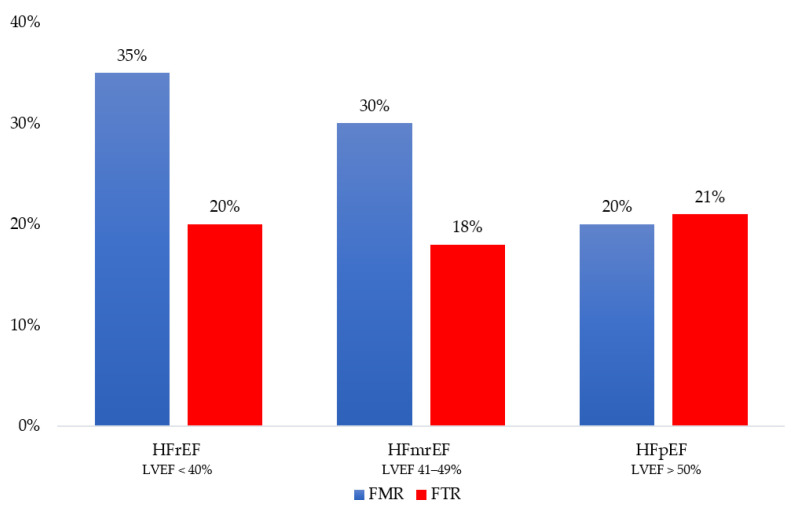
Distribution of functional mitral regurgitation (FMR) and functional tricuspid regurgitation (FTR) across the heart failure phenotype as defined by left ventricular ejection fraction (LVEF): reduced (HFrEF), mildly reduced (HFmrEF) and preserved (HFpEF).

**Figure 2 jcm-12-03316-f002:**
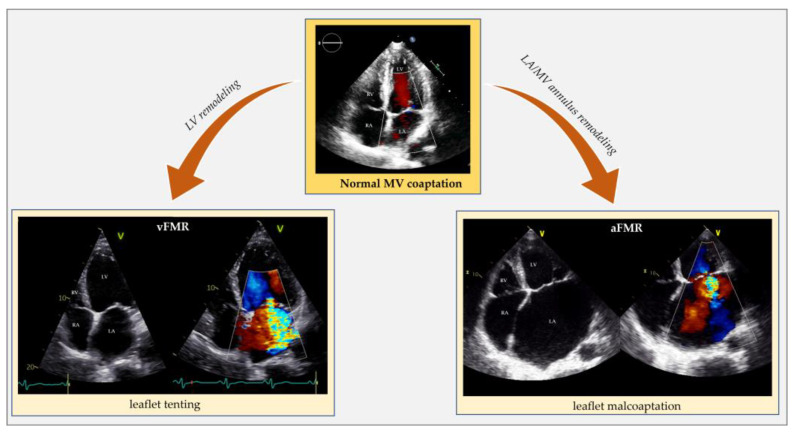
Echocardiographic comparison of normal mitral valve coaptation and functional mitral regurgitation in the context of left ventricle remodeling and dysfunction (vFMR) opposed to left atrial and annular dilatation (aFMR). aFMR: atrial functional mitral regurgitation; LA: left atrium; LV: left ventricle; MV: mitral valve; RA: right atrium; RV: right ventricle; vFMR: ventricular functional mitral regurgitation.

**Figure 3 jcm-12-03316-f003:**
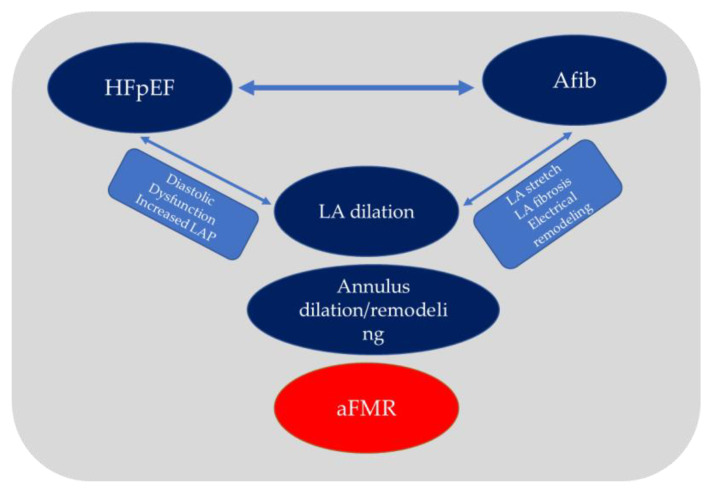
Pathophysiology of aFMR resulting from a sequential relationship between HFpEF (sinus rhythm) and atrial fibrillation. Adapted with permission from Deferm S. et al. [19].

**Figure 4 jcm-12-03316-f004:**
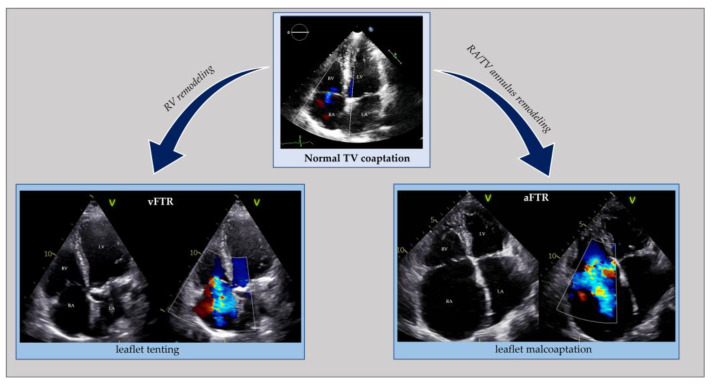
Echocardiographic comparison of normal tricuspid valve coaptation and functional tricuspid regurgitation in the context of right ventricle remodeling (vFTR) opposed to right atrial and annular dilatation (aFTR). aFTR: atrial functional tricuspid regurgitation; LA: left atrium; LV: left ventricle; RA: right atrium; RV: right ventricle; vFTR: ventricular functional mitral regurgitation; TV: tricuspid valve.

**Table 1 jcm-12-03316-t001:** Overview of transcatheter mitral valve repair and replacement devices, with CE approval, for treating mitral regurgitation (Adapted with permission from [64,65]).

Type of Intervention	Target Structure	Device	Description	Eligibility Criteria
Edge-to-edge	Mitral leaflets	MitraClip (Abbott Vascular, Abbott Park, IL, USA) PASCAL (Edwards Lifesciences, Irvine, CA, USA)	Based on edge-to-edge technique Transfemoral transeptal approach Approved for FMR and DMR	Central A2-P2 (ideal) No calcification Mean gradient < 4 mmHg MVA > 3 cm^2^ Sufficient leaflet tissue for grasping
Direct Annuloplasty	Mitral annulus	Cardioband (Edwards Lifesciences, Irvine, CA, USA)	Implantation of a flexible ring into the posterior annulus Ideal for annular dilatation mainly due to LA enlargement (atrial FMR) Anchoring on the hinge of the annulus Transfemoral transeptal approach	Annular dilatation with functional (or mixed, functional-dominant) etiology
Indirect Annuloplasty	Coronary sinus	Carillon (Cardiac Dimensions, Kirkland, WA, USA)	Nitinol anchors placed in the distal and proximal coronary sinus Reduction of MV annulus diameter upon deployment of the device Transjugular approach	Annular dilatation with functional (or mixed, functional-dominant) etiology Coronary sinus proximity and coplanarity
Chordal replacement	Papillary muscles	NeoChord (NeoChord, St Louis Park, MN, USA)	Surgical off-pump procedure Implantation of artificial chords Transapical access	Prolapse or flail Leaflet-to-annulus index ≥ 1.25
MV replacement	MV apparatus	Tendyne (Abbott Vascular, Abbott Park, IL, USA)	Self-expanding valve Indicated in suboptimal anatomy for transcatheter repair Transapical approach	MVA 1.0–3.0 cm^2^ Multisegment disease Commissural disease, perforations, clefts Mean gradients 5–10 mmHg Unlikely LVOT obstruction LVEF ≥ 30% Suboptimal MR reduction expected with transcatheter repair No scar or remodeled LV (transapical access)

DMR: degenerative mitral regurgitation; FMR: functional mitral regurgitation; LA: left atrium; LV: left ventricle; LVEF: left ventricle ejection fraction; LVOT: left ventricle outflow tract; MR: mitral regurgitation; MV: mitral valve; MVA: mitral valve area.

**Table 2 jcm-12-03316-t002:** Similarities and differences among COAPT, MITRA-FR, and CLASP trials with respect to study design and endpoints (Adapted with permission from [68,70]).

	COAPT	MITRA-FR	CLASP	CLASP (FMR)
Patients enrolled	614	304	124	85 (single arm)
Technical implantation success	98%	96%	96%	96%
Atrial fibrillation/Flutter	57.3%	34.5%	53.4%	45%
LVEF	31 ± 9%	33 ± 7%	44 ± 14%	37 ± 10%
EROA	41 ± 15 mm^2^	31 ± 10 mm^2^	38 ± 15 mm^2^	34 ± 11 mm^2^
LVEDV	101 ± 34 mL/m^2^	135 ± 35 mL/m^2^	181 ± 61 mL	199 ± 59 mL
Mortality at 1 y and 2 y	19% and 29%	23% and 34%	9% and 20%	12% and 28%
MR ≥ 3+ at discharge → 12 mo → 24 mo	7.4% → 5% → 0.9%	8% → 17% → not recorded	4% * → 0% → 3%	4% * → 0% → 5%

FMR: functional mitral regurgitation; LVEF: left ventricle ejection fraction; EROA: effective regurgitant orifice area; LVEDV: left ventricle end-diastolic volume. * Data at 30 days.

**Table 3 jcm-12-03316-t003:** Overview of transcatheter tricuspid valve repair and replacement devices, with CE approval, for treating functional tricuspid regurgitation (Adapted with permission from [82,83]).

Type of Intervention	Target Structure	Device	Description	Eligibility Criteria
Edge-to-edge	Tricuspid leaflets	TriClip (Abbott Vascular, Abbott Park, IL, USA) PASCAL (Edwards Lifesciences, Irvine, CA, USA)	Based on edge-to-edge technique Approximation of the septal and anterior leaflets or septal and posterior leaflets	Small septolateral gap ≤ 7 mm Anteroseptal jet location Trileaflet morphology Diffusely degenerated leaflets and pacemaker lead impingement are unfavorable anatomic conditions
Direct Annuloplasty	Tricuspid annulus	Cardioband (Edwards Lifesciences, Irvine, CA, USA)	Implantation of a flexible ring with multiple anchors on the hinge of the annulus Challenging procedure Distance between RCA and annulus may be a limitation	Annular dilatation as primary mechanism of TR Mild tethering (tenting height <0.76 cm, tenting area < 1.63 cm^2^, tenting volume < 2.3 mL) Central jet location Sufficient landing zone for anchoring
Heterotopic replacement	Superior and inferior caval veins	TricValve (Orbus Vienna AU, Wien, Austria)	Self-expanding valves Indicated in patients with significant backflow in the IVC and/or SVC Palliative care in unfavorable anatomy for transcatheter repair Irrespective of the TR etiology	Appropriate caval diameters (and intercaval distance) Contraindicated in severe RV dysfunction and pulmonary hypertension

FTR: functional tricuspid regurgitation; IVC: inferior vena cava; RA: right atrium; RCA: right coronary artery; RV: right ventricle; SVC: superior vena cava; TR: tricuspid regurgitation.

## Data Availability

Not applicable.

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
