# Peer review of "Functional Mitral and Tricuspid Regurgitation across the Whole Spectrum of Left Ventricular Ejection Fraction: Recognizing the Elephant in the Room of Heart Failure"

_jcm, 2023, doi:10.3390/jcm12093316_

Round 1
Reviewer 1 Report
Comments to Authors
Introduction
N/A
Epidemiology
Please consider illustrating the positive relationship between prevalence of FMR and decreasing LVEF and lack of relationship between prevalence of FTR and decreasing LVEF with a line graph for easier visualization of the concept.
Pathophysiology and Prognosis
“
Ventricular FMR typically occurs in HFrEF due to ischemic or non-ischemic ventric-ular disease.
“
What are the common non-ischemic ventricular diseases that cause vFMR in HFrEF?
“
FMR negatively impacts survival, either in HFpEF [adjusted hazard ratio (adj. HR) 1.40, 95% confidence interval (CI) 1.09–1.81; p = 0.009] [22], HFmrEF (adj. HR 1.72, 95% CI 1.24–2.39; p = 0.0012) [8] and HFrEF (adj. HR 1.61, 95% CI 1.22–2.12; p = 0.001] [2].
“
Should “FMR” be “vFMR”?
“
Atrial FMR is common in atrial fibrillation (AFib) but also occurs in sinus rhythm.
“
You throughly talk about the pathophysiology of a-fib causing aFMR. Can you please also talk about how sinus rhythm can lead to aFMR?
“
In a study by Dziadzko V et al., patients with ventricular FMR had the most signifi-cant LV remodeling, highest pulmonary pressure, lowest LVEF, stroke volume, and E/e'. Patients with atrial FMR presented smaller LV size, generally normal LVEF and stroke volume, with a modest MR volume and orifice, while E/e' and pulmonary pressure were elevated [33].
“
Didn’t patients with aFMR, compared to patients with vFMR, have larger LA size? If yes, please add it.
“
In advanced LA and LV remodeling, a net distinction between the atrial and ventricular mechanisms is no longer possible because these entities usually coexist.
“
Can’t imaging, such as TTE, TEE, or cardiac MRI, distinguish between atrial and ventricular mechanisms?
“
The prognosis of ventricular FMR is significantly worse than atrial FMR, and each etiology leads to different treat.
“
Is there any data on whether the relationship between vFMR and mortality is direct (i.e., primarily due to valvulopathy) or indirect (i.e., due to underlying comorbidities, such as ischemic heart disease)?
“
As for the left heart, right ventricular remodeling causing leaflet tethering and sys-tolic restricted motion is typical of ventricular FTR.
“
Please replace “left” with “right.”
Please list causes of RV remodeling that result in vFTR.
Therapeutic Implications
Regarding FMR, is there any data on the direct impact of GDMT, CRT, and surgical/transcatheter treatment on FMR?
Similarly, regarding FTR, is there any data on the direct impact of diuretics and sodium and water re-striction and surgical/transcatheter annuloplasty on FTR?
Transcatheter Therapy
Consider creating a table briefly summarizing COAPT, MITRA-FR, and CLASP for easier understanding of these three major trials.
Author Response
Rev #1
Epidemiology
Please consider illustrating the positive relationship between prevalence of FMR and decreasing LVEF and lack of relationship between prevalence of FTR and decreasing LVEF with a line graph for easier visualization of the concept.
Thank you for your suggestion. A new figure was added accordingly.
Pathophysiology and Prognosis
“Ventricular FMR typically occurs in HFrEF due to ischemic or non-ischemic ventricular disease.“
What are the common non-ischemic ventricular diseases that cause vFMR in HFrEF?
Thank you for your essential comment that allowed us to expand the pathophysiology of non-ischemic ventricular disease causing FMR in the setting of HFrEF. A paragraph was added.
“FMR negatively impacts survival, either in HFpEF [adjusted hazard ratio (adj. HR) 1.40, 95% confidence interval (CI) 1.09–1.81; p = 0.009] [22], HFmrEF (adj. HR 1.72, 95% CI 1.24–2.39; p = 0.0012) [8] and HFrEF (adj. HR 1.61, 95% CI 1.22–2.12; p = 0.001] [2].“
Should “FMR” be “vFMR”?
Thank you for your suggestion, but the data refer to the functional form of MR, which includes both the ventricular form (vFMR, more representative of HFrEF) and the atrial form (aFMR, more representative of HFpEF and HFmrEF). Currently, few data are available on the different forms of MR in the different spectrum of HF.
“Atrial FMR is common in atrial fibrillation (AFib) but also occurs in sinus rhythm.“
You throughly talk about the pathophysiology of a-fib causing aFMR. Can you please also talk about how sinus rhythm can lead to aFMR?
Thank you for your suggestion, which allows us to expand our pathophysiological discussion about the genesis of atrial FMR in sinus rhythm. Sinus rhythm is not directly the cause of aFMR, but aFMR can occur in HFpEF with increased LA pressure and LA remodeling, irrespective of atrial fibrillation. A new figure adapted from the paper of Deferm S. et al was added (copyright permission was obtained).
“In a study by Dziadzko V et al., patients with ventricular FMR had the most significant LV remodeling, highest pulmonary pressure, lowest LVEF, stroke volume, and E/e'. Patients with atrial FMR presented smaller LV size, generally normal LVEF and stroke volume, with a modest MR volume and orifice, while E/e' and pulmonary pressure were elevated [33].“
Didn’t patients with aFMR, compared to patients with vFMR, have larger LA size? If yes, please add it.
The LA size did not significantly differ between the three groups enrolled (vFMR, aFMR, and organic MR), being 54 ± 17, 52 ± 16 and 50 ± 18 mL/m2, respectively (p= 0.12). Following your suggestion, this has been clarified in the text.
“In advanced LA and LV remodeling, a net distinction between the atrial and ventricular mechanisms is no longer possible because these entities usually coexist.“
Can’t imaging, such as TTE, TEE, or cardiac MRI, distinguish between atrial and ventricular mechanisms?
We strongly agree that multimodality imaging can help distinguish the two forms of MR (aFMR vs vFMR) and provide functional and anatomic findings of the mitral valve, left ventricle, and atrium. Nevertheless, this sentence refers to the natural history of the disease, in which, at an advanced stage, all components negatively affect each other, feeding the vicious cycle of valve disease and HF. The same is true for tricuspid regurgitation, right atrium, and ventricle.
“The prognosis of ventricular FMR is significantly worse than atrial FMR, and each etiology leads to different treat. “
Is there any data on whether the relationship between vFMR and mortality is direct (i.e., primarily due to valvulopathy) or indirect (i.e., due to underlying comorbidities, such as ischemic heart disease)?
Thank you for your comment. This is a very interesting and contradictory point. Several and differing data are currently present, leading to confusion regarding FMR, its assessment, grading, interpretation and impact on the outcome. However, this issue has been addressed in the reviewed version of the manuscript.
“As for the left heart, right ventricular remodeling causing leaflet tethering and sys-tolic restricted motion is typical of ventricular FTR. “ Please replace “left” with “right.”
Thank you for your comment. We corrected the mistakes by changing the sentence
Please list causes of RV remodeling that result in vFTR.
A paragraph was added
Therapeutic Implications
Regarding FMR, is there any data on the direct impact of GDMT, CRT, and surgical/transcatheter treatment on FMR?
Thank you for your comment. These data are extensively reported in the numerous studies listed in the current guidelines for managing VHD and treating acute and chronic heart failure. However, in accordance with your question, we have expanded the paragraph addressing this issue.
Similarly, regarding FTR, is there any data on the direct impact of diuretics and sodium and water re-striction and surgical/transcatheter annuloplasty on FTR?
Again, thank you for your comment. Several studies support these recommendations, and the guidelines for VHD and HF endorse these therapeutic approaches. However, the revised version of the manuscript addresses this issue.
Transcatheter Therapy
Consider creating a table briefly summarizing COAPT, MITRA-FR, and CLASP for easier understanding of these three major trials.
Thank you for your suggestion. A new table has been added
Reviewer 2 Report
This is a general review of FMR and FTR focused on epidemiology, pathophysiology, prognosis and treatment. The article is well written, figures and tables are informative.
Intoduction (1) and Epidimiology (2) are well written sections of the article. Are there any data regarding the co-existence of the two pathologies (FMR and FTR)?
Pathophysiology (3) gives an nice general overview of the basic mechanisms. However , regarding the FMR , it is important to present and explain the concept of "proportionate and dispoportionate FMR" that has also an impact on treatment options. Describing ischemic and non ischemic mechanisms of FMR is also important. As for FTR, the mechanism of "ventricular FTR" is not sufficiently presented and the authors rapidly progress to present atrial FTR. This should be adressed.
Therapeutic implications section (4) and transcatheter therapy sections (also 4) are overlaping and I would suggest create two new sections : one for FMR treatment and one for FTR treatment. Presenting some key factors for CRT responders and non responders might be of interest. For FTR transcather treatment , the new percutaneous valves arriving in the market should be mentioned.
Eligibility criteria for percutaneous FMR and FTR correction should be presented more in detail.
Author Response
REV #2
This is a general review of FMR and FTR focused on epidemiology, pathophysiology, prognosis and treatment. The article is well written, figures and tables are informative.
Intoduction (1) and Epidimiology (2) are well written sections of the article. Are there any data regarding the co-existence of the two pathologies (FMR and FTR)?
Thank you for your essential comment. It is a very interesting point. The paragraph has been changed accordingly.
Pathophysiology (3) gives an nice general overview of the basic mechanisms. However , regarding the FMR , it is important to present and explain the concept of "proportionate and dispoportionate FMR" that has also an impact on treatment options.
Thank you for your comment. Although this statement sounds convincing, no conclusive data currently supports this issue. The concept of "disproportionality" is attractive and elegant from an intellectual standpoint. Nevertheless, it seems to be mainly a theoretical assumption, and there are serious methodological limitations in terms of calculating LV volumes and characterizing patient hemodynamics (Hagendorff A, Doenst T, Falk V. Echocardiographic assessment of functional mitral regurgitation: opening Pandora's box? ESC Heart Fail. 2019 Aug;6(4):678-685. doi: 10.1002/ehf2.12491)
Describing ischemic and non ischemic mechanisms of FMR is also important.
According to your comment and reviewer #1, the paragraph has been modified, expanding the pathophysiological description of ischemic and non-ischemic MR
As for FTR, the mechanism of "ventricular FTR" is not sufficiently presented and the authors rapidly progress to present atrial FTR. This should be adressed.
Thank you for your valuable suggestion, similar to reviewer #1. A paragraph was added accordingly.
Therapeutic implications section (4) and transcatheter therapy sections (also 4) are overlaping and I would suggest create two new sections : one for FMR treatment and one for FTR treatment. Presenting some key factors for CRT responders and non responders might be of interest. For FTR transcather treatment , the new percutaneous valves arriving in the market should be mentioned.
Thank you for your valuable comment. In accordance with your suggestion, we create two new subsections under the “therapeutic implications” section. We also added more data on CRT, TEER (Triluminate trial) and a mention of TR replacement.
Eligibility criteria for percutaneous FMR and FTR correction should be presented more in detail.
According to your valuable suggestion, Tables 1 and 3 have been changed by adding a dedicated column.
Reviewer 3 Report
3. Pathophysiology and prognosis:
Line 67: *valve tenting can be symmetric or asymmetric
Lines 72-75: Consider changing wording of these 2 sentences to: There is an exponential mortality increase for any EROA increment above a threshold of 0.10 cm2.
I do not understand Tables 1 and 2. Are these clinical characteristics derived from other studies? It appears these values are obtained from small studies that may not be representative of the entire population and so I do not see the utility of having them.
Would expand a little on the paragraph that mentions the worse prognosis in ventricular than atrial FMR and the need to distinguish between them as they have different prognostic and therapeutic implications. I think its an important paragraph.
Line 110: *As for the right side of the heart..
Lines 123-131 should be 1 paragraph
4. Therapeutic implications:
In the table showing TMVR therapies, isn't NeoChord a surgical procedure? Does it belong to that table?
There are 2 tables labeled '3'
Author Response
REV #3
- Pathophysiology and prognosis:
Line 67: *valve tenting can be symmetric or asymmetric .
Thank you. It was revised.
Lines 72-75: Consider changing wording of these 2 sentences to: There is an exponential mortality increase for any EROA increment above a threshold of 0.10 cm2.
Thank you for your suggestion. The sentence was rephrased.
I do not understand Tables 1 and 2. Are these clinical characteristics derived from other studies? It appears these values are obtained from small studies that may not be representative of the entire population and so I do not see the utility of having them.
In accordance with your valuable comment, we deleted Tables 1 and 2. Some findings that we consider notable in the differences between the groups have been reported in the main text.
Would expand a little on the paragraph that mentions the worse prognosis in ventricular than atrial FMR and the need to distinguish between them as they have different prognostic and therapeutic implications. I think its an important paragraph.
Thank you for your comment. This is a very interesting and contradictory point. This issue has been addressed in the reviewed version of the manuscript.
Line 110: *As for the right side of the heart..
Thank you for your comment. We corrected the mistakes changing with the following sentence: “similarly to the left side of the heart”
Lines 123-131 should be 1 paragraph
The paragraph was correct
- Therapeutic implications:
In the table showing TMVR therapies, isn't NeoChord a surgical procedure? Does it belong to that table?
NeoChord
Thank you for the observation. The doubt of whether it is a surgical or an interventional procedure remains. However, since the NeoChord is a transcatheter transapical procedure echo-guided, it has been included in transcatheter mitral valve therapies (Yousef S, et al. Transcatheter mitral valve therapies: State of the art. J Card Surg. 2022;37:225-233).
There are 2 tables labeled '3'
Thank you. We corrected it.
Round 2
Reviewer 2 Report
Thank you for these corrections. Still I think that presenting the concept of proportionate and disproportionate MR would add some value in the FMR section.
Author Response
According to the reviewer's suggestion, we added a new paragraph about the concept of proportionate and disproportionate MR.